# Non-Pharmacological Preventive Measures Had an Impact on COVID-19 in Healthcare Workers before the Vaccination Effect: A Cohort Study

**DOI:** 10.3390/ijerph19063628

**Published:** 2022-03-18

**Authors:** Mireia Utzet, Fernando G. Benavides, Rocío Villar, Andrea Burón, Maria Sala, Luis-Eugenio López, Pau Gomar, Xavier Castells, Pilar Diaz, José María Ramada, Consol Serra

**Affiliations:** 1Centre for Research in Occupational Health, Department of Medicine and Life Sciences, Universitat Pompeu Fabra, 08003 Barcelona, Spain; fernando.benavides@upf.edu (F.G.B.); rvillar@psmar.cat (R.V.); pdiazperez@psmar.cat (P.D.); jramada@psmar.cat (J.M.R.); consol.serra@upf.edu (C.S.); 2IMIM-Hospital del Mar Medical Research Institute, 08003 Barcelona, Spain; aburon@psmar.cat (A.B.); msalaserra@psmar.cat (M.S.); xcastells@psmar.cat (X.C.); 3CIBER of Epidemiology and Public Health, 28029 Madrid, Spain; 4Occupational Health Service, Parc de Salut Mar, 08003 Barcelona, Spain; 5Department of Epidemiology and Evaluation, Parc de Salut Mar, 08003 Barcelona, Spain; 6Network for Research on Chronicity, Primary Care and Health Promotion (RICAPPS), 08003 Barcelona, Spain; 7Consorci Mar Parc de Salut de Barcelona, 08003 Barcelona, Spain; lelopezmartin@psmar.cat (L.-E.L.); pgomar@psmar.cat (P.G.)

**Keywords:** COVID-19, healthcare worker, incidence, rate ratio, non-pharmaceutical preventive measures

## Abstract

Healthcare workers have been and still are at the forefront of COVID-19 patient care. Their infection had direct implications and caused important challenges for healthcare performance. The aim of this study is to assess the impact of non-pharmacological preventive measures against COVID-19 among healthcare workers. This study is based on a dynamic cohort of healthcare workers (*n* = 5543) who had been hired by a Spanish hospital for at least one week during 2020. Negative binomial regression models were used to estimate the incidence rate and the rate ratio (RR) between the two waves (defined from 15 March to 21 June and from 22 June to 31 December), considering natural immunity during the first wave and contextual variables. All models were stratified by socio-occupational variables. The average COVID-19 incidence rate per 1000 worker-days showed a significant reduction between the two waves, dropping from 0.82 (CI95%: 0.73–0.91) to 0.39 (0.35–0.44). The adjusted RR was 0.54 (0.48–0.87) when natural immunity was acquired during the first wave, and contextual variables were considered. The significant reduction of the COVID-19 incidence rate could be explained mainly by improvement in the non-pharmacological preventive interventions. It is needed to identify which measures were more effective. Young workers and those with a replacement contract were identified as vulnerable groups that need greater preventive efforts. Future preparedness plans would benefit from these results.

## 1. Introduction

In the middle of the worse pandemic in a century [1], Spain has been one of the most affected countries in the world in terms of absolute numbers of diagnosed cases and mortality due to the coronavirus disease (COVID-19) [1]. Like most European countries, Spain experienced a first wave between March and June 2020 (B.1 being the dominant variant, a large European lineage, the origin of which roughly corresponds to the Northern Italian outbreak early in 2020 [2]), which was flattened as a result of a radical lockdown, including non-essential services and school closures. From 20 May onwards, the mandatory use of masks was established. In June, the country gradually returned to relative normality, but the number of COVID-19 cases increased again, entailing the beginning of a second wave in July until December (when the variant B.1.177, major and mostly European lineage, was dominant [2]). This second wave was flattened mainly through non-pharmaceutical interventions at the community level, such as strict limitation of mobility and reduced cultural, entertainment and restaurant activities, although primary and secondary schools were open and most economic activities maintained [3].

Healthcare workers have been and still are at the forefront of COVID-19 patient care, developing their work in an environment where the probability of exposure to the virus was the highest due to daily exposure to confirmed or suspected COVID19 patients [4]. WHO estimates that, globally, health care workers have accounted for 10% of all reported cases of coronavirus and that between 80,000 and 180,000 may have died from COVID-19 in the period January 2020 to May 2021. In Spain, official reports estimate that about 24.1% (41,000 individuals) of confirmed cases from the first wave were healthcare workers [5]. A similar situation was observed in Italy [6] and Sweden [7], where up to 20% of the total confirmed cases were healthcare workers. This high proportion of cases among healthcare workers, probably lower than the actual number due to low availability of the polymerase chain reaction (PCR) test at the beginning of the pandemic, had direct implications and caused important challenges for health system performance and could have probably contributed to the acceleration of the pandemic and the overload of the health system [8]. However, before vaccinations at the end of December in the European Union [9], a significant decrease in the COVID-19 incidence trend was observed among healthcare workers in Spain during the second wave [10].

At the beginning of the pandemic, several non-pharmacological preventive measures were implemented in Spanish hospitals to limit nosocomial and community transmission. However, as in most European hospitals, there was a serious shortage of essential personal protective equipment (e.g., masks and gowns) [11]. In addition, HCWs could only be tested for severe acute respiratory syndrome coronavirus 2 (SARS-CoV-2) if they showed well-defined symptoms, and the action protocol changed over the months. It was not until mid-May that this situation began to be progressively stabilised so that PPE was fully available from July (during the second wave), and both the quarantine and isolation criteria and protocol changes were scarce and manageable. Vaccination, in combination with non-pharmaceutical interventions, is the best way to control the pandemic [12]. Due to their high risk of infection and their role in the nosocomial transmission of SARS-CoV-2 [13], HCWs have been a priority group for COVID-19 vaccine administration [14], which started in Spain in January 2021.

The aim of this study is to assess the impact of non-pharmacological preventive measures against COVID-19 among a healthcare workers cohort in Spain, comparing the first two waves of the pandemic during 2020 and considering their demographic and occupational characteristics.

## 2. Materials and Methods

### 2.1. Study Design and Population Size

This study is based on a dynamic cohort for the period from 1 January to 31 December 2020, including all healthcare workers (*n* = 5543) who had been hired by PSMar for at least one week during 2020.

Healthcare workers’ information was available from the Human Resources Department databases. For each healthcare worker, we retrieved socio-demographic and labour information, including sickness absence records (starting and ending dates). In addition, we obtained information about COVID-19 confirmed cases among those healthcare workers from the occupational epidemiological surveillance system (GO.DATA [15]) used by the Occupational Health and the Epidemiology Departments. To link both databases and to ensure confidentiality, a participant identification number for the study was created. Privacy and data safety were guaranteed, and the study was approved by the PSMar Ethical Committee (final approval on 9 October 2020).

### 2.2. Definitions and Variables Information

A COVID-19 case was defined as a healthcare worker with a confirmed diagnosis by a positive PCR test performed at PSMar (where the Abbott Alinity m SARS-CoV-2 assay [16] has been used during the whole pandemic). The number of effective work days per worker was estimated by summing up the number of days according to their contracts and subtracting the number of days of sick leave per worker attributable to COVID-19 (isolation or quarantine) or other diseases. The first wave was defined according to the Spanish lockdown period, between 15 March and 21 June, and the second from 22 June to 31 December [17].

For each worker, we included information about type of contract (permanent, temporary, and replacement); occupational category (physicians, nurses and nurse aides, other healthcare workers, such as medical and other trainees or lab technicians and administration and management staff); and centre within PSMar, which are Hospital del Mar (acute care), Hospital de l’Esperança (acute care), Centre Fòrum (long term care and psychiatry), Centre Dr. Emili Mira (psychiatry), and others. Socio-demographic variables were also included: sex and age (18–29 years, 30–49 years, and 50–70 years). Finally, two contextual variables were included: the COVID-19 incident cases in Barcelona city per week [18], and the number of newly hospitalised patients attributable to SARS-CoV-2 infection per week at PSMar (data provided by the epidemiology and evaluation service of PSMar).

The cohort sample size was 5543 workers, of which 4066 were women (73.4%) with a median age of 37 years (IQR 27–49); 3421 (61.7%) had a permanent contract; 51.1% were nurses or aides and 15.1% physicians; 3749 (67.7%) worked at Hospital del Mar (see Appendix A).

### 2.3. Statistical Analysis

A first univariate analysis was carried out, estimating for each week and for both waves the incidence rate (IR) per 1000 worker-days and its 95% confidence intervals (CI95%), according to socio-demographic and occupational variables. The IR was defined as the ratio of the COVID-19 cases divided by the effective work days. Second, the rate ratio (RR) between the two waves was estimated by negative binomial regression, introducing a dummy variable into the model and identifying the period time (first and second wave), taking the first wave as a reference. The number of effective worked days by week per worker were included in the model as an offset. Four consecutive approaches were used: (1) fitting a crude baseline model, (2) excluding first-wave-infected healthcare workers from the second wave IR estimates, (3) including in the model the COVID-19 incident cases in Barcelona city per week, and (4) including in the model the number of newly hospitalised patients attributable to COVID-19 per week at PSMar. These four approaches were estimated stratifying each of the demographic and occupational variables. All the calculations were conducted with STATA version 14.2 (Stata Corp., College Station, TX, USA).

## 3. Results

The COVID-19 IR time trend during 2020 increased dramatically at the beginning of March (week 10), with an IR of 3.5 per 1000 worker-days at week 12 and decreased quickly after the first lockdown until an IR near 0 at week 19. This was the situation at PSMar until the end of the first wave, in week 25. The second wave, starting at week 26, showed a less dramatic increasing trend from week 40 to week 44 when it reached a peak IR of 1.2 per 1000 worker-days (Figure 1).

Figure 1 also shows the trend of COVID-19 incident community cases in Barcelona city and the number of newly hospitalised patients due to COVID-19 per week at PSMar. The number of community cases was higher in the second wave than during the first, with around 3000 cases at week 13 versus more than 6000 cases at week 42. Meanwhile, the number of new cases admitted at PSMar was around 550 cases in week 13 versus around 100 cases in week 43.

During the first wave, 333 new COVID-19 cases among healthcare workers were identified, with an average IR of 0.82 (CI95%: 0.73–0.91); during the second wave, with 311 new COVID-19 cases, the IR was 0.39 (0.35–0.44) (Table 1). In both waves, the average IR was higher among healthcare workers younger than 30 years old, those with a replacement work contract, and among nurses and aides.

The global RR (Table 2) showed a slight and non-significant increase from 0.48 (CI95%: 0.41–0.57) in Model 1 to 0.51 (CI95%: 0.43–0.60) in Model 2, in which infected healthcare workers during the first wave were excluded from the second wave IR estimates,; this was raised to 0.54 (CI95% 0.48–0.87) in Model 4 when both the number of community cases and the number of newly hospitalised patients attributable to COVID-19 at PSMar were included in the analysis. However, the RR was lowest in Model 3 when only the number of community cases was included: 0.24 (CI95%: 0.19–0.31).

When stratifying by the socio-demographic and occupational variables, looking at the saturated Model 4, the RR estimates presented differences by age, from a non-significant RR of 0.93 (CI95%: 0.60–1.42) among workers younger than 30 years down to a significant RR of 0.38 (CI95%: 0.24–0.60) and 0.33 (CI95%: 0.18–0.62) among those aged 30 and 50 years and those older than 50, respectively. By type of contract, a significant RR among permanent work contract workers of 0.43 (CI95%: 0.31–0.62) was observed; by centre, the RR ranged between 0.12 (CI95%: 0.04–0.37) for Hospital de l’Esperança and 0.67 (CI95%: 0.48–0.92) for Hospital del Mar; by occupational category, the RR was 0.31 (CI95%: 0.12–0.78) among physicians and 0.49 (CI95%: 0.35–0.69) among nurses and aides.

## 4. Discussion

Results showed a significant RR between the two first pandemic waves, revealing a substantially lower COVID-19 incidence rate among the healthcare workers of PSMar in the second wave, with noteworthy inequalities by sociodemographic and occupational characteristics. After adjusting for the number of COVID-19 community cases and the number of hospital admissions, both the main sources of health workers’ infection, the RR continued to be statistically significant.

Given this decrease, and without the action of the vaccines, the most plausible explanation could be the improvement of the non-pharmaceutical preventive measures implemented at PSMar. These include the availability of adequate PPE, the correct use of which reduces significantly the transmission of infection [19,20] and universal screening, i.e., testing both symptomatic and asymptomatic HCWs, which is vital for effective cases detection and, thus, for limiting transmission [20,21]. According to hospital documentation and the information from key informants (occupational risk prevention technicians and head of the logistics department of PSMar, personal communication, June 2021), the lack of equipment during the first weeks of the pandemic was gradually overcome from mid-April (week 17) onwards, stabilising the purchase and distribution of adequate PPE (mainly FPP2 and sanitary masks, eye protection and gowns) available to all healthcare workers by the end of June. In addition to performing PCR on symptomatic and asymptomatic HCWs, systematic screening started in August. This result suggests the maintenance of non-pharmacological preventive measures in hospitals, considering that, despite increased vaccination coverage, healthcare workers still need to protect themselves from the nosocomial transmission of SARS-CoV-2. Hence, future studies are needed to identify all the non-pharmaceutical preventive measures implemented and their specific effectiveness.

The impact of all these non-pharmaceutical measures could be probably higher if resources for testing had been higher at the beginning of the pandemic since the number of cases was probably higher [22]. Otherwise, acquired natural immunity by healthcare workers might have played a more modest role in the explanation of the difference between both periods. A final result, which we find remarkable in this “natural experiment”, is that during the second wave, there was an increase in community incidence of cases, mainly attributed to the resumption of travel and the relaxation of control and containment measures, rather than to the new variant (B.1.117), which appeared in Spain in early July [2], as it has not been shown to be more transmissible than the previous one [23]. This higher community incidence, contrasting with the lower incidence among healthcare workers, further underlines the important role played by non-pharmaceutical preventive measures at PSMar. Moreover, the opposite trend is observed in similar hospitals in Poland [24] and Germany [25], which show a significant increase in the incidence among healthcare workers, concluding that their preventive strategy should be improved. However, it should also be noted that the lower number of admissions of patients with COVID-19 in the second period compared to the first may have also contributed to the lower incidence of cases among healthcare workers.

On the other hand, the RR was higher in young workers, those with a replacement contract and those working at Hospital del Mar, where most infected patients were admitted; they were identified as vulnerable groups that needed greater efforts to improve non-pharmaceutical preventive measures. For instance, healthcare workers younger than 30 years experienced a higher incidence and a higher and non-significant RR. This result could be explained because younger workers are more frequently employed in temporary and replacement contracts (data not shown). This is consistent with the literature in occupational health that often shows that younger workers are more vulnerable because of more precarious employment conditions [26]. This finding might also be explained in terms of risk perception and adherence to preventive behaviours, which might be lower among younger people, as previously suggested [27], but should be thoroughly studied. Furthermore, the incidence among other healthcare workers, including medical and other trainees and younger workers with a temporary contract, confirms this cluster of vulnerability during the second wave in comparison with their colleagues.

Nurses and nurse assistants showed higher incidence rates during both the first and second waves. This result is consistent with other similar studies among healthcare workers [20,28,29,30] and reflects the different roles and working conditions between occupational categories. Furthermore, healthcare workers involved in the admission of patients may have an increased risk of infection [31], but as the PSMar protocol of admission did not change during 2020, we think that it does not help explain our results. Future research is needed to confirm this preliminary hypothesis. Nonetheless, the most exposed workers (nurses, aides and physicians) presented a low but statistically significant RR, while the less exposed (administration and management staff) showed a non-significant RR. This result supports the hypothesis that the improvements in non-pharmacological preventive measures were the main explanation for the decreasing incidence between the two waves.

Our study has some limitations. First, our study is based only on one hospital, and we need to compare our results in relation to other hospitals. Second, non-pharmacological preventive measures are considered in this study globally, and we need to assess the effectiveness of the selection of some of the non-pharmaceutical preventive measures. Third, incidence rates were calculated as an average over each period rather than their time trends. However, the analysis was to compare both periods. The current study also has several strengths. This study was based on real data before the vaccination effect. Data sources were reliable administrative and health data (already collected) to report relevant information on a key aspect of the pandemic [32]. To our knowledge, this is the first study that compares the COVID-19 incidence impact among HCWs between two periods of the pandemic, considering occupational characteristics as well as contextual factors.

## 5. Conclusions

In summary, our research highlights the potential impact of non-pharmaceutical preventive measures, which are key elements for an adequate preparedness plan of the health system to possible future health crises due to a new virus or any expression of the climate crisis (e.g., heat waves, floods). According to our results, proper planning of non-pharmaceutical preventive measures could have prevented almost half of the cases among healthcare workers.

## Figures and Tables

**Figure 1 ijerph-19-03628-f001:**
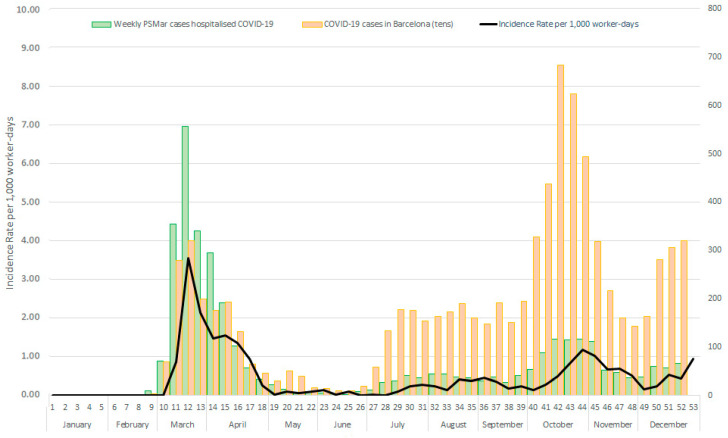
COVID-19 IR per 1000 worker-days among PSMar healthcare workers, hospitalised COVID-19 cases, and COVID-19 cases in Barcelona city by weeks.

**Table 1 ijerph-19-03628-t001:** Incident COVID-19 cases, effective working days, incidence rate (IR) per 1000 worker-days, during the first and second waves. By sex, age, contract arrangement, building centre, occupational category, and work shift. PSMar healthcare workers, 2020.

		First Wave			Second Wave		
		Incident Cases	Effective Days Worked	IR1 (CI95%)	Incident Cases	Effective Days Worked	IR2 (CI95%)
Sex	Women	238	297,688	0.80 (0.70–0.91)	220	580,883	0.38 (0.33–0.43)
	Men	95	109,584	0.87 (0.71–1.06)	91	208,603	0.44 (0.35–0.53)
Age	18–29	101	101,914	0.99 (0.81–1.20)	138	199,188	0.69 (0.58–0.82)
	30–49	136	188,793	0.72 (0.61–0.85)	117	361,673	0.32 (0.27–0.39)
	50–70	96	116,481	0.82 (0.67–1.00)	55	228,474	0.24 (0.18–0.31)
Contract arrengement	Permanent	229	307,541	0.74 (0.65–0.85)	177	581,040	0.30 (0.26–0.35)
	Temporary	42	46,778	0.90 (0.66–1.20)	26	51,617	0.50 (0.34–0.73)
	Replacement	62	52,857	1.17 (0.91–1.49)	108	156,829	0.69 (0.57–0.83)
Health building centres	Hospital Mar (acute care)	205	274,433	0.75 (0.65–0.85)	236	533,075	0.44 (0.39–0.50)
	Hospital Esperança (acute care)	56	39,803	1.41 (1.07–1.81)	24	71,565	0.34 (0.22–0.49)
	Fòrum centre (long term care, psychiatry)	25	28,802	0.87 (0.57–1.26)	17	54,035	0.31 (0.19–0.49)
	Dr. Emili Mira Centre (psychiatry)	45	38,286	1.18 (0.87–1.56)	28	80,520	0.35 (0.24–0.50)
Occupational category	Physicians	49	72,702	0.67 (0.50–0.88)	33	135,463	0.24 (0.17–0.34)
	Nurses and aides	219	206,572	1.06 (0.93–1.21)	186	409,975	0.45 (0.39–0.52)
	Other health-care workers	44	66,436	0.66 (0.49–0.88)	68	122,459	0.56 (0.43–0.70)
	Administration and management staff	21	61,562	0.34 (0.22–0.51)	24	121,589	0.20 (0.13–0.29)
Total		333	407,272	0.82 (0.73–0.91)	311	789,486	0.39 (0.35–0.44)

**Table 2 ijerph-19-03628-t002:** Rate ratio (95%CI) of incident COVID19 cases between waves among PSMar healthcare workers, 2020. Crude and adjusted negative binomial regression models by sex, age, contract arrangement, health building centre, and occupational category.

		Model 1	Model 2	Model 3	Model 4
	Women	0.48 (0.39; 0.59)	0.50 (0.41; 0.61)	0.23 (0.17; 0.30)	0.53 (0.38; 0.74)
	Men	0.49 (0.36; 0.68)	0.52 (0.38; 0.71)	0.28 (0.18; 0.43)	0.57 (0.35; 0.94)
Age	18–29 years	0.72 (0.54; 0.95)	0.75 (0.57; 0.99)	0.45 (0.32; 0.64)	0.93 (0.60; 1.42)
	30–49 years	0.44 (0.33; 0.58)	0.46 (0.35; 0.61)	0.19 (0.13; 0.28)	0.38 (0.24; 0.60)
	50–70 years	0.29 (0.20; 0.42)	0.31 (0.22; 0.45)	0.11 (0.07; 0.20)	0.33 (0.18; 0.62)
Contract arrangement	Permanent	0.41 (0.33; 0.51)	0.43 (0.35; 0.53)	0.20 (0.14; 0.27)	0.43 (0.31; 0.62)
	Temporary	0.59 (0.35; 0.98)	0.61 (0.36; 1.01)	0.28 (0.14; 0.58)	0.64 (0.30; 1.36)
	Substitution	0.60 (0.42; 0.85)	0.63 (0.44; 0.89)	0.34 (0.22; 0.52)	0.84 (0.47; 1.51)
Centre	Hospital Mar (acute care)	0.58 (0.48; 0.72)	0.62 (0.50; 0.75)	0.33 (0.25; 0.43)	0.67 (0.48; 0.92)
	Hospital Esperança (acute care)	0.26 (0.16; 0.44)	0.28 (0.17; 0.47)	0.06 (0.03; 0.14)	0.12 (0.04; 0.37)
	Fòrum centre (long term care, psychiatry)	0.44 (0.22; 0.88)	0.47 (0.24; 0.93)	0.20 (0.07; 0.58)	0.67 (0.22; 2.04)
	Dr. Emili Mira Centre (psychiatry)	0.24 (0.14; 0.43)	0.26 (0.15; 0.46)	0.09 (0.04; 0.21)	0.36 (0.15; 0.85)
Occupational category	Physician	0.34 (0.20; 0.57)	0.36 (0.22; 0.60)	0.13 (0.06; 0.28)	0.31 (0.12; 0.78)
	Nurses and nurse assistants	0.42 (0.34; 0.52)	0.45 (0.36; 0.55)	0.21 (0.15; 0.28)	0.49 (0.35; 0.69)
	Other health-care workers	0.94 (0.61; 1.44)	0.96 (0.63; 1.48)	0.54 (0.32; 0.92)	1.03 (0.55; 1.94)
	Administration and management staff	0.59 (0.32; 1.12)	0.60 (0.32; 1.13)	0.35 (0.16; 0.78)	0.58 (0.22; 1.51)
Total		0.48 (0.41; 0.57)	0.51 (0.43; 0.60)	0.24 (0.19; 0.31)	0.54 (0.48; 0.87)

Model 1: crude model. Model 2: Model 1, excluding first wave positive cases in the second wave IR estimation. Model 3: Model 2, including COVID-19 cases in Barcelona. Model 4: Model 3, including new COVID-19 hospitalisation cases.

## Data Availability

The datasets generated and/or analysed during the current study are available from the corresponding author upon reasonable request.

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
