# Peer review of "Non-Pharmacological Preventive Measures Had an Impact on COVID-19 in Healthcare Workers before the Vaccination Effect: A Cohort Study"

_ijerph, 2022, doi:10.3390/ijerph19063628_

Round 1

Reviewer 1 Report

The present study hypothesizes that the most plausible explanation for the decrease in incidence could be the improvement of the non-pharmaceutical preventive measures implemented in the PSMar. The idea is good. Similar results were obtained in the studies of A Zurochka (2021), who explained the lower seropositivity rates in high-risk persons and HCWs by the use of PPI. With that, A Hildebrandt (2022) observed the increase in seroprevalence of seroprevalence from the first to third wave. Please expand the discussion section.

  1. In order to make this hypothesis more convincing, will you be so kind to find the data on the timing of the receipt of personal protective equipment in the hospital, and the degree of its availability. When you can link this to the dynamics of the incidence, it will be a good confirmation of your hypothesis.
  2. Please give the information, what changes were made in the protocol of admission of patients in these terms (recording by phone, single window mode ...That point could greatly affect the incidence rates.
  3. Moreover, please compare the observed dynamics with the dynamics on the levels of city and country
  4. It will be great if you discuss the role of dominant (at those time-points) kinds of SARS-CoV2 strains, as they had various transmissibility (EB Hodcroft, 2021, CDC). Moreover, you can add the additional lines on the Figure 1.
  5. Please give information about the PCR tests used (manufacture, sensitivity).

Overall, the study is interesting, thank you.

Hodcroft EB, Zuber M, Nadeau S, Vaughan TG, Crawford KHD, Althaus CL, Reichmuth ML, Bowen JE, Walls AC, Corti D, Bloom JD, Veesler D, Mateo D, Hernando A, Comas I, González-Candelas F; SeqCOVID-SPAIN consortium, Stadler T, Neher RA. Spread of a SARS-CoV-2 variant through Europe in the summer of 2020. Nature. 2021 Jul;595(7869):707-712. doi: 10.1038/s41586-021-03677-y. Epub 2021 Jun 7. PMID: 34098568.

Zurochka A, Dobrinina M, Zurochka V, Hu D, Solovyev A, Ryabova L, Kritsky I, Ibragimov R, Sarapultsev A. Seroprevalence of SARS-CoV-2 Antibodies in Symptomatic Individuals Is Higher than in Persons Who Are at Increased Risk Exposure: The Results of the Single-Center, Prospective, Cross-Sectional Study. Vaccines (Basel). 2021 Jun 9;9(6):627. doi: 10.3390/vaccines9060627. PMID: 34207919; PMCID: PMC8229032.

Sonmezer MC, Erul E, Sahin TK, Rudvan Al I, Cosgun Y, Korukluoglu G, Zengin H, Telli Dizman G, Inkaya AC, Unal S. Seroprevalence of SARS-CoV-2 Antibodies and Associated Factors in Healthcare Workers before the Era of Vaccination at a Tertiary Care Hospital in Turkey. Vaccines (Basel). 2022 Feb 8;10(2):258. doi: 10.3390/vaccines10020258. PMID: 35214715; PMCID: PMC8875971.

Hildebrandt A, Hökelekli O, Uflacker L, Rudolf H, Paulussen M, Gatermann SG. Seroprevalence of SARS-CoV-2 Antibodies in Employees of Three Hospitals of a Secondary Care Hospital Network in Germany and an Associated Fire Brigade: Results of a Repeated Cross-Sectional Surveillance Study Over 1 Year. Int J Environ Res Public Health. 2022 Feb 19;19(4):2402. doi: 10.3390/ijerph19042402. PMID: 35206589; PMCID: PMC8878380.

Korona-GÅ‚owniak I, Mielnik M, Podgajna M, Grywalska E, Hus M, Matuska K, Wojtysiak-Duma B, Duma D, Glowniak A, Malm A. SARS-CoV-2 Seroprevalence in Healthcare Workers before the Vaccination in Poland: Evolution from the First to the Second Pandemic Outbreak. Int J Environ Res Public Health. 2022 Feb 17;19(4):2319. doi: 10.3390/ijerph19042319. PMID: 35206504; PMCID: PMC8871845.

Reviewer 2 Report

Dea Editor

The article by Mireia Utzetet al , entitlet : “Non-pharmacological preventive measures impact against 2 COVID-19 on healthcare workers, before the vaccination effect: 3 a cohort study” is a more interest Article focused on the impact of non-pharmacological preventive measures against COVID-19 among healthcare workers. .Autors reported the data of a dynamic cohort of healthcare 5,543 workers. They reported that the young workers and those with a replacement contract were identified as vulnerable groups that need greater preventive efforts.

The paper needs thorough revision to be understandable and conclusive

  • Please delect : It is needed to identify which measures were more effective, in the abstract section
  • Please spell out abbreviations such as “SARS-CoV2” and “COVID-19” when appear first in the main text.
  • Please expand the introduction section with the data on COVID-19 and Healthcare workers, and on COVID vaccination
  • Pease add in the conclusion section a commet on the non resporer to COVID-19 Vaccination

The document opens a very interesting scenario in the problem of the COVID 19 healthcare setting. 

I believe that the document is valid after minor revision for publication without further amendments.
